# Impact of Lockdown Measures on Health Outcomes of Adults with Type 2 Diabetes Mellitus in Bangladesh

**DOI:** 10.3390/healthcare11081191

**Published:** 2023-04-21

**Authors:** Farhana Akter, Ahsanul Haq, Brian Godman, Kona Chowdhury, Santosh Kumar, Mainul Haque

**Affiliations:** 1Department of Endocrinology, Chittagong Medical College Hospital, Chattogram 4203, Bangladesh; 2Infectious Diseases Division, icddr, b, Mohakhali, Dhaka 1212, Bangladesh; 3Strathclyde Institute of Pharmacy and Biomedical Sciences, University of Strathclyde, Glasgow G4 0RE, UK; 4Division of Public Health Pharmacy and Management, School of Pharmacy, Sefako Makgatho Health Sciences University, Pretoria 0208, South Africa; 5Centre of Medical and Bio-Allied Health Sciences Research, Ajman University, Ajman 346, United Arab Emirates; 6Department of Pediatrics, Gonoshasthaya Samaj Vittik Medical College, Dhaka 1344, Bangladesh; 7Department of Periodontology and Implantology, Karnavati School of Dentistry, Karnavati University, Gandhinagar 382422, Gujarat, India; 8Unit of Pharmacology, Faculty of Medicine and Defence Health, Universiti Pertahanan Nasional Malaysia (National Defence University of Malaysia), Kem Perdana Sungai Besi, Kuala Lumpur 57000, Malaysia

**Keywords:** Type 2 Diabetes Mellitus, COVID-19, medical records, symptom control, lockdown, Bangladesh

## Abstract

COVID-19 lockdown measures appreciably affected patients’ lifestyles, negatively impacting on their health. This includes patients with Type 2 Diabetes Mellitus (T2DM). Care of these patients was also negatively impacted due to a priority to treat patients with COVID-19, certainly initially, within hospitals and clinics in Bangladesh, combined with a lack of access to clinics and physicians due to lockdown and other measures. This is a concern in Bangladesh with growing rates of T2DM and subsequent complications. Consequently, we sought to critically analyze the situation among patients with T2DM in Bangladesh during the initial stages of the pandemic to address this information gap and provide future direction. Overall, 731 patients were recruited by a simple random sampling method among patients attending hospitals in Bangladesh, with data collected over 3 timescales: before lockdown, during the pandemic, and after lockdown. Data extracted from patients’ notes included current prescribed medicines and key parameters, including blood sugar levels, blood pressure, and comorbidities. In addition, the extent of record keeping. The glycemic status of patients deteriorated during lockdown, and comorbidities as well as complications related to T2DM increased during this period. Overall, a significant proportion of key datasets were not recorded in patients’ notes by their physician before and during lockdown. This started to change after lockdown measures eased. In conclusion, lockdown measures critically affected the management of patients with T2DM in Bangladesh, building on previous concerns. Extending internet coverage for telemedicine, introduction of structured guidelines, and appreciably increasing data recording during consultations is of the utmost priority to improve the care of T2DM patients in Bangladesh.

## 1. Introduction

Several public health measures were introduced across various countries to reduce the spread of COVID-19, and its subsequent impact on morbidity and mortality, in the absence of proven effective treatments and vaccines [1,2,3]. The measures included lockdown and social distancing, wearing of personal protective equipment (PPE), closure of schools and universities, transport restrictions, as well as the introduction of tracing and quarantining measures [3,4,5,6,7,8,9,10,11]. Bangladesh was no exception, with multiple activities introduced at the start of the pandemic to slow the spread of the virus [12,13,14]. There were also major changes in hospitals to reduce or prevent potential transmission from healthcare staff to patients, and vice versa, as well as from patient to patient. This was especially important as there can easily be 5000 patients a day with outpatient appointments in some public hospitals. The changes included reducing the number of clinics or their closure, with physicians seeking to limit personal contact with patients to reduce nosocomial transmission, and similarly for some patients [15,16]. As a result, there was an increase in telemedicine and other approaches to manage patients, especially those with non-communicable diseases (NCDs), during the pandemic, which helped to replace in-person visits [17,18,19,20].

Moves to online consultations can be a concern, however, since such changes may lead to social isolation and loss of regular check-ups, potentially negatively impacting on adherence to medicines and necessary lifestyle changes. Alongside this, there are concerns with the affordability, knowledge, and the infrastructure associated with telemedicine, especially among patients in low- and middle-income countries (LMICs) [18,19,20,21,22]. This can be a major issue for patients with diabetes, especially those in LMICs [23].

Diabetes is a priority area since it is one of the most prevalent NCDs globally, and there are severe clinical manifestations if patients are not carefully managed, which can result in increased morbidity, mortality, and costs [24,25,26]. This includes Bangladesh, where the prevalence of diabetes, including T2DM, has been growing in recent years, and now affects up to 11.8% of the population [27,28]. These growing prevalence rates have also increased mortality, morbidity, and health-related costs in Bangladesh, exacerbated by concerns with managing coronary vascular disease (CVD) in these patients [29,30,31,32]. This is especially the case among the elderly in Bangladesh, where it has been estimated that nearly 60% of the elderly have at least one NCD, including diabetes, with nearly one-quarter having multiple comorbidities [33]. The increased prevalence of diabetes is of the greatest concern in urban areas, including Dhaka and Chittagong in Bangladesh [28], with up to ten million patients receiving care for their diabetes across Bangladesh before the first wave of the pandemic [34]. Overall, it is estimated that in Bangladesh, the number of patients with diabetes will
increase to 13.7 million by 2045, unless addressed [29]. Of equal concern is that this may be an under-estimate considering the current appreciable under-diagnosis of patients with diabetes in Bangladesh, exacerbated by the recent COVID-19 pandemic and access to clinics [22,33,35].

Similar to other chronic comorbid conditions, managing patients with T2DM during COVID-19 lockdown can be difficult. As a result, potentially, increases in weight with changes in diet and exercise, as well as reducing control over blood sugar levels, can occur, with non-diabetic individuals found to have an increased risk of T2DM due to weight gain because of lockdown measures [36,37,38,39]. However, this may not always be the case [38,40,41,42,43,44]. Having said this, various published studies have shown concerns with weight gain, the control of blood sugar levels, and potentially an increase in complications among patients with T2DM during the pandemic, which have been exacerbated by issues of compliance with prescribed medicines [36,45,46,47]. In addition, people with diabetes have been found to be more susceptible to being infected with COVID-19, especially those with an uncontrolled glycemic status. If the persistence of poor glycemic control increases among individuals with diabetes, more people will become prone to infections, which will lead to an increase in caseloads [23,47].

We believe few studies to date have been undertaken to assess the impact of COVID-19, and associated lockdown measures, on the management of patients with T2DM in Bangladesh and the future implications, although this is changing [48]. This is a concern considering, as mentioned, the increasing rates of T2DM in Bangladesh [29]. Alongside this, there are also concerns with the high rates of patient co-payments, rising prices, and issues of affordability with managing patients with diabetes in Bangladesh, which have been exacerbated by employment concerns during the pandemic through lockdown and other measures [13,27,28,35,49].

Consequently, we sought to address this information gap with the long-term objective to reduce the extent of complications among patients with T2DM, with their associated impact on morbidity and costs, in this and future pandemics. Our initial pilot study among eight patients in Chittagong Medical College showed that there was typically poor control of blood glucose levels of patients with T2DM during the pandemic [50]. However, there was better control of blood pressure, lipids, and albumin levels. There were also concerns with the appreciable extent of missing knowledge gaps in patients’ notes, especially during the pandemic. In view of this, the aim here was to undertake a substantial study in this and other hospitals in Bangladesh. As a result, we could provide practical approaches that can be used to improve the management of patients with T2DM in Bangladesh during this and future pandemics. This includes measures to enhance the continuity of care as well as improve the communication with patients, caregivers, and healthcare providers during a pandemic.

## 2. Materials and Methods

A mixed-methods approach was used. This included a cross-sectional study among patients with T2DM attending the Department of Endocrinology, Chittagong Medical College Hospital, Chittagong, Bangladesh, a tertiary care public hospital, and Chevron Clinical Laboratory and Diagnostic Center, between December 2019, i.e., before the start of the COVID-19 pandemic, and November 2021. When the clinic at Chittagong Hospital is fully functioning, each physician in the outpatient clinics will typically see 120 to 150 patients in a morning session (lasting from 8.00 am until 1.00 pm), with the hospital typically seeing 5000 patients a day. The patient numbers are lower in the Endocrinology clinics, at 40 to 50 patients per session. The patient numbers during a session are also lower in private clinics, including Chevron Clinical Laboratory and Diagnostic Center. Patients typically attend private clinics in the afternoon, contrasting with clinics in public hospitals which usually operate in the morning.

In Bangladesh, both public and private hospitals have not yet entered digitalized data in a preserving system. These research data were obtained from the patients themselves, preserved or maintained in their treatment file, which they retain. The principal researcher had no control over whether a patient appropriately conserved all records, both for prescription and laboratory findings. Consequently, the possibility for missing data increased as the whole process was manually maintained. Moreover, during the COVID-19 pandemic, families were typically disturbed; hence, treatment files were possibly not well-maintained.

These two hospitals, one public and one private, were purposely chosen to provide data for this study, with an earlier study in Chittagong Medical College Hospital documenting high prevalence rates of T2DM among patients attending outpatient clinics in the hospital [31]. In addition, there is a high prevalence of diabetes in Chittagong, Bangladesh [28]. The inclusion of both public and private hospitals in this study reflects the current situation of managing these patients in Bangladesh [51,52].

### 2.1. Generation of Case Report Forms

A specifically designed case report form (CRF) was used to assess key measures regarding the management of patients with T2DM. This included HbA1c levels, fasting plasma glucose (FPG) and postprandial blood glucose (PPG) levels, blood pressure (BP), lipid levels (HDL, LDL, triglycerides, and total cholesterol), serum creatinine, and urinary albumin, as well as a range of oral and injectable medicines prescribed to control glycemic levels among T2DM patients, including insulins. The extent of any micro- or macro-vascular complications, as well as any other relevant complications or comorbidities, including thyroid disorders, were also captured in the CRF. The final content of the CRF was based on published papers combined with the knowledge of the co-authors, and subsequently tested in a pilot study before full rollout [31,32,50,53,54]. The clinical datasets included in the CRFs, potential targets, and their references are included in Appendix A.

### 2.2. Study Design

Retrospective data were taken from patients’ notes. Baseline data were taken from before the start of the COVID-19 pandemic and the subsequent initiation of lockdown measures in Bangladesh, as well as during the pandemic in four divided periods, six months apart. The time periods were: December 2019 (Visit 1), June 2020 (Visit 2), December 2020 (Visit 3), June 2021 (Visit 4), and November 2021 (Visit 5). A six-monthly interval was seen as optimal to monitor the changes in key clinical measures [50].

All diabetic patients attending the Endocrinology outpatient department of Chittagong Medical College and at the Chevron Clinical Laboratory and Diagnostic Center that came to consult their physician were potentially recruited into the study. Typically, patients attending public hospitals in Bangladesh are comparatively poorer and less educated compared with those attending private clinics. Educational background is intimately related to patients’ awareness of their disease as well as preserving key documents. This is important in Bangladesh, with, as mentioned, patients typically carrying their documents with them rather than residing in any central repository within hospitals.

Exclusion criteria included those patients without any notes as well as pregnant patients.

At the beginning of the pandemic, due to the lockdown measures, patients could not visit hospitals or physician clinics. In addition, many private physician clinics were also closed. Public hospitals remained open during this period; however, they were typically only dealing with patients with COVID-19, and not those with diabetes. As a result, routine follow-up visits for patients with diabetes were dramatically reduced. Alongside this, patients were also afraid of visiting hospitals due to the fear of COVID-19 infection. The situation had eased by the end of 2021, even though the pandemic was not over. However, its virulence was decreasing.

Since the COVID-19 pandemic was ongoing during the study period with the initiation of lockdown measures, including social distancing, patients typically only started visiting clinics again in November 2021 when they were recruited into the study. Once recruited, a retrospective analysis was undertaken of their accompanying documents. This is because, as mentioned, patients typically carry their own notes with them, with no central repository within the hospitals. In addition, detailed questioning of their medication adherence and any problems, alongside routine monitoring of key datasets, during face-to-face contact with physicians, especially in the public sector, is difficult. This especially considering the number of patients with diabetes seen in a morning clinic in public hospitals and the specific areas that need to be discussed to help control blood sugar levels as well as prevent complications.

There were no sample size calculations since the evidence of the impact of lockdown and other measures on the management of patients with T2DM is conflicting [36,37,41,42,43].

The developed questionnaire included demographic data, comorbidities, medication compliance, and daily habits before and after the lockdown was introduced. This built on the clinical data requirements contained in the CRFs (Appendix A), building on the pilot study [50].

### 2.3. Statistical Analysis

Descriptive statistical analyses were performed using categorical variables with numbers and proportions. All statistical analyses were performed using Stata/MP 15.0, and a machine learning technique was used to perform sentimental analysis to evaluate the most common complaints of the studied participants using Python. Sentiment analysis, also known as opinion mining, is a technique in natural language processing that involves using machine learning algorithms to identify and extract subjective information from textual data, e.g., common complaints from patients. Specifically, this technique involves analyzing a given text to determine the emotional tone (diseases) or sentiment. Machine learning is used in sentiment analysis to train algorithms to recognize patterns in language that indicate a particular sentiment. The process usually involves two phases: training and testing. In the training phase, a dataset of labeled text is used to train the algorithm to recognize specific sentiment patterns. In the testing phase, the algorithm is applied to new, unlabeled text data, and its performance is evaluated based on how well it can correctly identify the sentiment of the text.

To ascertain lipids and cardiometabolic markers in different periods of the COVID-19 pandemic, a univariate regression model was conducted, and figures were prepared using GraphPad Prism 8.3.2. A *p*-value of <0.05 was considered significant.

No attempt was made to revise the data if patient records were deficient in a number of parameters. This is because we wanted to document the actual situation to provide future guidance, knowing that record keeping, especially during the pandemic, was suboptimal, with both patients and physicians seeking to minimize the time spent in hospital clinics.

### 2.4. Ethical Approval

Ethical approval was obtained from the Institutional Review Board of Chittagong Medical College, 57, KB Fazlul Kader Road, Police Station: Panchlaish, Post Office: Chittagong 4203, Bangladesh, obtained with Reference No.: CMC/PG/2021/232, Dated: 6 November 2021. Written informed consent was sought from patients before data collection. Consent was entirely voluntary, with no pressure placed on patients to comply.

## 3. Results

### 3.1. Patients and Their Demographics

A total of 730 patients were recruited for the study following verbal consent, with Table 1 describing their characteristics. Among the recruited patients, 504 (69.0%) were female and 226 (31.0%) were male, with almost all the patients being married. From the available information, the levels of obesity were higher among female patients (38.6%) versus males (19.7%).

### 3.2. Key Clinical Data

#### 3.2.1. Clinical Characteristics and Treatments

Table 2 discusses the regular follow-up of patients during the study period. This includes details of extensive missing data before and during the early stages of the pandemic. The situation had appreciably changed by Visits 4 and 5.

Table 3 documents details of the key clinical features of the patients that were recorded and available in their notes between December 2019 and November 2021, with an appreciable number not recorded until their visit in November 2021.

The lack of recording of patients’ details in Table 2 and Table 3 reflects the fact that typically there is only limited time for consultations with patients during their clinic visit. This occurs due to high patient volumes, especially in public hospitals, and the need to quickly review each patient’s notes at the start of the short consultation.

Both hospitals taking part in the study are referral hospitals and take patients from both rural and urban areas. Consequently, not every clinical detail will be discussed with patients during each consultation. This reflects the current reality among patients with diabetes in Bangladesh, especially in the public system, with the situation worsening as prevalence rates grow. In addition, both physicians and patients felt the need to reduce any contact time during the COVID-19 pandemic, thereby keeping any consultation and record keeping to a minimum. However, this began to change by Visit 5, which resulted in appreciably greater follow-up of patients and recording of their key clinical details.

Figure 1 depicts differences in key parameters regarding plasma glucose and HbA1c levels, as well as key cardiovascular markers including blood pressure, before, during, and following the easing of lockdown measures. The figures show the mean alongside the standard deviation, and a univariate regression model was used to estimate the *p*-values. There was a significant increase in FPG (*p* = 0.035), PPG, and HbA1c levels (*p* = 0.009) once COVID-19 restrictions had been eased (Visit 5) compared to pre-pandemic levels or mid-pandemic (Visit 1). There was also an increase in a number of lipid levels from the middle visits to Visit 5, i.e., TC (*p* = 0.006), LDL (*p* = 0.005), and TG (*p* = 0.029). However, there was a decrease in LDL levels following the initiation of lockdown measures; although, there was an increase after this. There were no changes in either systolic or diastolic blood pressure recordings during the pandemic. However, the data for the initial visits must be treated with caution in view of appreciable missing data. This has been exacerbated by the fact that test data can easily be lost when patients are responsible for their own notes, especially in the public system, and physicians do not always record patient details in their notes due to time pressures. This was further exacerbated during the pandemic, with both patients and physicians wanting to keep contact time, as well as time in the hospital, to a minimum.

Appendix A records details of oral anti-diabetic medicines and insulins prescribed to patients over the past years to help control their blood sugar levels, with Appendix A providing further details of the injectable medicines prescribed, including insulins.

All patients were prescribed oral anti-diabetic medicines to help control their blood glucose levels, which were supplemented by injectable medicines including insulins if needed (Appendix A). However, this was not always recorded in the patients’ notes. Patients do, however, need a prescription from a physician to be able to obtain their oral medicines or insulin from a community pharmacist.

There was typically no discussion with patients regarding their medicine taking, especially when no medicines were recorded. Consequently, some patients may follow the instructions from previous medications and some may reduce the number or doses of medicines they take themselves, especially if they are experiencing side effects and there was no contact with physicians during the pandemic. Some patients may also be dispensed different medicines from community pharmacists on request, and some may just stop taking their medicines all together due to side effects and other parameters.

Appendix A documents the range of medicines prescribed to help control patients’ blood pressure, coagulation, and angina, with Appendix A documenting details of the lipid-lowering medicines prescribed and recorded. These medicines are important to help minimize the extent of cardiovascular events arising from T2DM.

#### 3.2.2. Complications and Comorbidities

Appendix A provide details of the extent of complications and comorbidities seen among the studied patients, with Appendix A providing additional details.

The principal reasons for hospitalization during the study period included issues of infection and cardiovascular disease, as well as waiting for surgery for the complications associated with diabetes.

Common complaints among the patients, when recorded, were documented via the sentimental approach using machine learning. Most patients had hypertension (HTN), hypothyroidism, obesity, non-alcoholic fatty liver disease (NAFLD), and ischemic heart disease (IHD).

## 4. Discussion

We believe this is the first study published in Bangladesh that monitors the actual management of patients with diabetes attending two referral centers in a leading city in Bangladesh, before and during the recent COVID-19 pandemic, as well as when lockdown restrictions were eased. This is important as there have been concerns with patient management across countries if there is limited access to treatment centers during the pandemic, and patients with T2DM are not being encouraged to continually improve their lifestyles as well as adhere to their prescribed medicines [22,45,46,55,56,57]. In addition, the high prevalence of uncontrolled T2DM and hypertension, as well as delays in the diagnosis of cardiovascular diseases among patients with diabetes before and during the pandemic, with its associated lockdown measures [29,30,31,32], further exacerbates concerns in this high-priority population [48,50,58,59]. We are aware that the pandemic has also adversely impacted on the diagnosis and management of other NCDs in LMICs [60,61]. This needs to be taken into consideration in future pandemics as a counter-consideration to aggressive lockdown measures [60,61].

There continued to be concerns with the level of control of blood glucose levels during Visits 1 to 5 (Table 3), with key parameters worsening from Visit 3 onwards (Figure 1). However, there was a considerable lack of recording of key details in patients’ notes in Visit 1, which carried on with limited patient contact until Visit 5. Consequently, it is difficult to comment on any changes made in the prescribing of both oral medicines and insulin to better control blood sugar levels. Encouragingly, in Visit 5, there was considerably more recording of key clinical parameters, which included the medicines prescribed. However, there is still appreciable room for improvement with a considerable lack of recording of any insulins prescribed to further improve the management of patients with T2DM in Bangladesh. This is important if patients with T2DM are to be appropriately managed, which is a growing concern in Bangladesh with increasing prevalence rates [27,28,48].

Encouragingly, blood pressure levels and lipid levels (total cholesterol) documented in Table 3 appeared to be within the agreed levels when these were recorded in patients’ notes (Appendix A). However, a worsening of lipid levels was observed towards the end of the pandemic, where this information was more thoroughly documented. This needs urgent addressing as cardiovascular events are a key complication of patients with diabetes if their diabetes is not properly managed [24]. Appendix A shows limited prevalence rates for both micro- and macro-vascular complications in the studied patients. However, these rates will increase if patients’ diabetes remain uncontrolled and their blood pressure and lipid levels do not remain within target levels (Appendix A). This will also impact on subsequent infection and hospitalization rates (Appendix A). Consequently, these key clinical parameters need improved monitoring in future visits.

Patient comorbidities (Appendix A) are as expected for this population. The most common comorbidity was hypothyroidism, with its highest frequency at the last visit, similar to hyperthyroidism but at a lower frequency. Bronchial asthma and COPD were more prevalent from the third visit onwards, especially around the end of the lockdown period. It is difficult to say anything further considering the extent of missing data in patients’ notes.

Potential ways forward to improve record keeping among patients with T2DM in Bangladesh include the use of electronic health records or other eHealth approaches. However, this may take time in Bangladesh considering the limited activities to date [62]. Telehealth as well as eHealth approaches, including remote monitoring of blood glucose levels as well as key parameters including blood pressure, can potentially address the multiple challenges experienced during the current and future pandemics. As a result, we can seek to improve the monitoring of patients with T2DM to reduce potential complications and their impact. However, again, this will take time to become established in Bangladesh, especially among patients attending public clinics [62,63]. mHealth approaches can also increase interactions between patients and physicians to enhance their motivation for lifestyle changes and adherence to medicines. This is especially important among patients living in rural areas, where access to specialist clinics can be difficult [64,65], and during lockdown, which severely impacted patients attending clinics in Bangladesh and across LMICs.

These are considerations for the future. In the meantime, there is a need to enhance record keeping among T2DM patients attending ambulatory care clinics in hospitals throughout Bangladesh, building on the improvements recorded in Visit 5 in these two hospitals. There are already ongoing activities in these two hospitals to improve record keeping in patients’ notes to be able to improve future patient management, and we will be following this up in future studies. This will complement activities by the government in Bangladesh, which is also planning steps to improve the recording of patient information by supplying logistics support and trained manpower in public hospitals.

We are aware of a number of limitations with our study. Firstly, we only collected data from two hospitals; however, these were carefully selected to cover both public and private hospitals. There were also considerable gaps in patients’ records, especially during the pandemic in order to avoid prolonged exposure time between physicians and patients as well as between patients in busy clinics. In addition, patients themselves missed clinic appointments due to the fear of catching COVID-19 and wanted to keep clinic time to a minimum, which subsequently impacted on the extent of information being recorded in their notes. There were also concerns regarding prescriptions and whether these were collected or taken to pharmacies for filling during the pandemic. Finally, patients themselves may have lost their records during the pandemic. This has created considerable bias in the findings. However, despite these concerns, we believe our findings reflect the current reality for patients with T2DM in Bangladesh and do provide guidance on ways to improve the future care of these patients. We will be following up our findings as lockdown pressures ease and concerns with the management of these patients in Bangladesh persist. This includes potential ways to improve record keeping.

## 5. Conclusions

In conclusion, patients with T2DM need regular follow-up to achieve glycemic targets as well as prevent delays in the complications associated with poor control of blood sugar levels. This is a concern during the pandemic, alongside the poor documentation in general of the clinical characteristics of patients in Bangladesh. Telemedicine and other approaches can help address concerns with patient contact and follow-up during a pandemic, especially around key areas of lifestyle modifications and adherence to prescribed medicines. However, challenges remain. Governments and other health organizations should collaborate, firstly to fully evaluate the impact of the pandemic on the management of patients with T2DM across sectors considering the identified challenges. Subsequently, this collaboration could help develop robust guidance to manage patients with T2DM during any pandemic, including the follow-up and motivation of patients, as well as ensuring a regular supply of medicines to treat their diabetes and subsequent complications. This is important in Bangladesh considering the rising prevalence rates of T2DM and its impact. Alongside this, data preservation is also critical to improve the management of patients and their outcomes, which includes managing patients to meet the agreed upon target levels. Consequently, governments in LMICs should focus on electronic health records or other eHealth approaches in the future as the prevalence of NCDs continues to rise. It is worth mentioning that the government of Bangladesh has already taken initiatives to better conserve patient data and records in a computerized model, and we will be following this up. We hope that the private hospitals of the country will follow the same path of developing a digitalized data-keeping system to improve patient care. Therefore, we can anticipate that overall, healthcare research and clinical outcomes will be improved.

## Figures and Tables

**Figure 1 healthcare-11-01191-f001:**
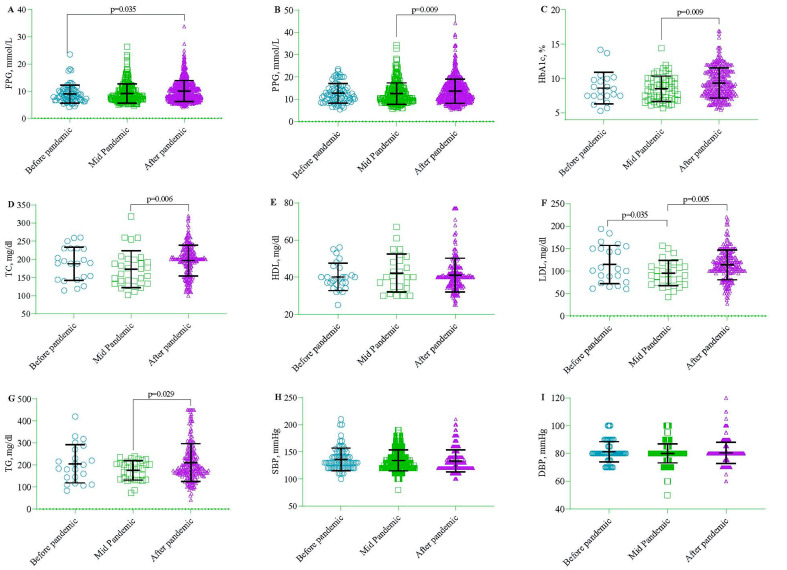
Key diabetic, lipids, and cardiometabolic markers in different periods of the COVID-19 pandemic. FPG = fasting plasma glucose levels (**A**); PPG = post prandial plasma glucose (**B**); HbA1c% (**C**); TC = total cholesterol (**D**); HDL = high-density lipoprotein (**E**); LDL = low-density lipoprotein (**F**); TG = triglyceride (**G**); SBP = systolic blood pressure (**H**); DBP = diastolic blood pressure (**I**).

**Table 1 healthcare-11-01191-t001:** Demographic characteristics of the patients.

Variables	Number (% in Brackets)
Age, mean ± SD	51.2 ± 11.3
**Age, Category**	
<30 years	31 (4.20%)
31–50 years	322 (44.1%)
>50 years	377 (51.6%)
**Sex**	
Male	226 (31.0%)
Female	504 (69.0%)
**Marital Status**	
Married	728 (99.7%)
Unmarried	2 (0.30%)
**Obesity (based on available information of 565 patients)**	
Male	27 (19.7%)
Female	165 (38.6%)

**Table 2 healthcare-11-01191-t002:** Details of regular follow-up and clinic attendance between December 2019 and November 2021 (n = 730).

	Visit 1	Visit 2	Visit 3	Visit 4	Visit 5
Regular Follow-Up					
Yes	61 (8.4%)	14 (2.0%)	26 (3.8%)	42 (6.1%)	510 (69.9%)
No	6 (0.80%)	8 (1.2%)	53 (7.7%)	259 (35.5%)	194 (26.6%)
Not recorded	663 (90.8%)	708 (97.0%)	651 (89.2%)	429 (58.8%)	26 (3.6%)

Note: NR = not recorded.

**Table 3 healthcare-11-01191-t003:** Key clinical findings between December 2019 and November 2021.

	Visit 1	Visit 2	Visit 3	Visit 4	Visit 5
**BMI**	25.43 ± 11.41	26.25 ± 8.83	29.14 ± 2.04	28.4 ± 4.80	28.1 ± 4.81
NR	721 (98.8%)	706 (96.7%)	678 (92.8%)	445 (61.0%)	128 (17.5%)
**Uncontrolled DM**					
Yes	41 (5.6%)	11 (1.5%)	24 (3.3%)	44 (6.0%)	293 (40.1%)
No	26 (1.2%)	11 (1.5%)	55 (7.5%)	257 (35.2%)	293 (40.1%)
NR	663 (90.8%)	708 (90.8%)	708 (90.7%)	429 (58.8%)	27 (3.7%)
**Blood sugar levels**					
FPG, mmol/L	10.0±2.04	10.75 ± 3.37	11.5±±4.7	16.07 ± 45.3	20.5 ± 46.0
PPG, mmol/L	19.2 ± 23.5	13.85 ± 4.00	10.0 ± 2.04	12.6 ± 4.85	17.5 ± 8.3
HbA1c, %	7.82 ± 1.52	8.28 ± 2.75	8.62 ± 2.29	8.51 ± 1.85	9.75 ± 1.85
**Lipid levels**					
TC, mg/dL	208.7 ± 30.16	208.2 ± 16.77	189.6 ± 49.33	172.1 ± 51.7	196.4 ± 45.3
HDL, mg/dL	42.14 ± 2.54	42.40 ± 6.19	40.15 ± 7.36	42.24 ± 10.28	41.9 ± 14.7
LDL, mg/dL	120.4 ± 28.47	121.2 ± 19.58	114.9 ± 42.43	102.0 ± 37.8	113.8 ± 33.22
TG, mg/dL	177.9 ± 49.56	217.6 ± 46.86	205.5 ± 86.43	186.2 ± 61.2	218.5 ± 119.0
**Blood pressure and creatinine levels**					
SBP, mmHg	128.7 ± 20.10	137.6 ± 22.87	136.3 ± 19.3	134.2 ± 19.3	133.2 ± 20.2
DBP, mm Hg	81.52 ± 8.45	85.0 ± 11.42	81.16 ± 7.24	80.03 ± 6.80	80.4 ± 7.62
Serum creatinine mg/dL	0.86 ± 0.07	0.74 ± 0.09	0.76 ± 0.72	1.01 ± 0.72	1.00 ± 0.61

Note: NR = not recorded; FPG = fasting plasma glucose levels; PPG = post prandial plasma glucose; TC = total cholesterol; HDL = high-density lipoprotein; LDL = low-density lipoprotein; TG = triglyceride; SBP = systolic blood pressure; DBP = diastolic blood pressure.

## Data Availability

Additional data are available upon reasonable request from the corresponding authors.

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
