# Peer review of "Impact of Lockdown Measures on Health Outcomes of Adults with Type 2 Diabetes Mellitus in Bangladesh"

_healthcare, 2023, doi:10.3390/healthcare11081191_

Round 1

Reviewer 1 Report (Previous Reviewer 1)

Thank you for revising the manuscript according to my suggestions.

A few minor modifications are still necessary, please see below.

Title: I would skip "the interferences" and think of something like "Impact of COVID-19 lockdown measures on health outcomes of adults with Type 2 Diabetes Mellitus in Bangladesh"

Introduction: I did not find the heading Introduction at the beginning of the introduction

lines 32-33: "face-to face interviews once lockdown measures eased" Does it mean that interviews were carried out only during "visit 5"? 

lines 112-113 "in this hospital". I understand that the first study was carried out in one hospital, but this used the data of two hospitals. So perhaps it is correct to say "in these hospitals".

English is better but it still needs to be further revised and edited.

Author Response

Comments and Suggestions for Authors

1) Thank you for revising the manuscript according to my suggestions.

Author comments: Thank you for your help – very much appreciated!

2) A few minor modifications are still necessary, please see below.

Author comments: Thank you – we hope we have now adequately addressed these.

3) Title: I would skip "the interferences" and think of something like "Impact of COVID-19 lockdown measures on health outcomes of adults with Type 2 Diabetes Mellitus in Bangladesh"

Author comments: Thank you – now changed.

4) Introduction: I did not find the heading Introduction at the beginning of the introduction

Author comments: Thank you – now made clear in the original revised paper – now hopefully OK.

5) lines 32-33: "face-to face interviews once lockdown measures eased" Does it mean that interviews were carried out only during "visit 5"? 

Author comments: Thank you – now updated and made clearer in the Abstract as in reality the patients were rarely interviewed because of time pressures, with the data taken from their notes. We hope this clarifies the situation.

6) lines 112-113 "in this hospital". I understand that the first study was carried out in one hospital, but this used the data of two hospitals. So perhaps it is correct to say "in these hospitals".

Author comments: Thank you – now done.

7) English is better but it still needs to be further revised and edited

Author comments: Thank you – now further updated. We hope this is now acceptable.

Reviewer 2 Report (Previous Reviewer 2)

Authors made a good effort and impovec their manuscript. However there still some points need improvement

1. Abstract should have the form "introduction- materials methods- results-conclusion"

2. Line 55 they should change "nosocomial transmissions" to "admission to hospitals"

3. Line 167 they should change " fear of catching COVID 19" to "with the fear of COVID 19 infections"

4. Although there was of fear even in doctors  in line 230 the excuse that doctors did not keep records and reduced time for patients is not acceptable as an excuse. Maybe authors should rephrase

5. Line 259 even if it was not recorded that patients took medicines doctors should ask this question otherwise it is missing data

6. In discussion authors just refer to results already analyzed previously. Maybe they should compare their study with other similar studies or studies interfering with other diseases.

Author Response

Comments and Suggestions for Authors

1) Authors made a good effort and impovec their manuscript. However there still some points need improvement

Author comments: Thank you for this – appreciated! We hope we now adequately address the remaining comments.

  1. Abstract should have the form "introduction- materials methods- results-conclusion"

Author comments: Thank you for this. Unfortunately – this is the regulations from the Journal. We hope this is OK with you.

  1. Line 55 they should change "nosocomial transmissions" to "admission to hospitals"

Author comments: Thank you – now made clearer. We hope this is now OK.

  1. Line 167 they should change " fear of catching COVID 19" to "with the fear of COVID 19 infections"

Author comments: Thank you – now changed.

5. Although there was of fear even in doctors  in line 230 the excuse that doctors did not keep records and reduced time for patients is not acceptable as an excuse. Maybe authors should rephrase

Author comments: Thank you – we have now updated this to give more details as to why this situation has occurred in Bangladesh. We hope this is now clearer.

6. Line 259 even if it was not recorded that patients took medicines doctors should ask this question otherwise it is missing data

Author comments: Thank you for this. However – as indicated this is not always possible in very busy clinics – exacerbated by the pandemic. We have now made this clearer, and hope this is now acceptable

7. In discussion authors just refer to results already analyzed previously. Maybe they should compare their study with other similar studies or studies interfering with other diseases.

Author comments: Thank you. We have now added in further references from across different countries demonstrating similar issues in patients with diabetes/ hypertension. We trust this is now OK.

Reviewer 3 Report (Previous Reviewer 3)

Table 1 from line 210 onwards requires a reference to the S1 table in the appendix in the Obesity column, stating that the WHO classification for the Asian population is used for the definition, as the values from Table 3 with BMI would be implausible. The Obesity line could, for example, contain BMI >27.5 and the reference so that it is clear that these values differ significantly from the North American classification.

In line 213 it should read Table 2 and not 3.

In Table 2, the missing record rate for Visit 1 is 90.8%. Later it is argued that the documentation rate gets worse under the pandemic, which is not evident from visits 2 and 3. Shouldn't we rather assume that overall the recording was critically poor before the pandemic as well? The text states ka that the values from Visit 1 come from a pre-study and thus from the time before the pandemic.

Line 248: The explanation of the abbreviations from Table 2 should also be under Figure 1, as the figures should be understandable by themselves.

Line 279/280: The sentence: "Common complaints among the patients, when recorded, were documented via the sentimental approach using machine learning" needs explanation. What exactly is meant by 'sentimental approach using machine learning'? Please explain which parameters were recorded or evaluated in this regard.

Line 326 better: Given the extent of missing data in the patients' records, it is difficult to give further details and interpretations on this.

Line 374 ff: As a reviewer, I rather believe that no guidelines should be developed for monitoring diabetes during a pandemic, but that the measures taken during the pandemic should now be critically evaluated, as there must be reasonable doubt that the measures taken were all reasonable, appropriate and effective.

Author Response

Comments and Suggestions for Authors

  1. Table 1 from line 210 onwards requires a reference to the S1 table in the appendix in the Obesity column, stating that the WHO classification for the Asian population is used for the definition, as the values from Table 3 with BMI would be implausible. The Obesity line could, for example, contain BMI >27.5 and the reference so that it is clear that these values differ significantly from the North American classification.

Author comments: Thank you – you are correct. We used Asian BMI's used to calculate obesity. For your information, Asian BMIs are different from e.g. American counterparts because of the difference in body fat and musculature. Asian people tend to have more body fat, especially abdominal fat, which accounts for a higher risk of developing T2DM and CVD. Consequently, for a given BMI, Asian people tend to have a higher weight related disease risk at lower BMIs than for instance their American counterparts. For Americans, BMI, 18.5 -24.9 is Healthy, ≥ 25 to 29.9 is seen as overweight, ≥30 kg/ m2 is considered as obese. For our population in Bangladesh, as indicated in Table S1, the normal range is 18.50 – 22.99; overweight: ≥23.00; and obese: ≥27.50. We have also added a new reference to this effect, and hope everything is now OK.

  1. In line 213 it should read Table 2 and not 3.

Author comments: Thank you – now changed

  1. In Table 2, the missing record rate for Visit 1 is 90.8%. Later it is argued that the documentation rate gets worse under the pandemic, which is not evident from visits 2 and 3. Shouldn't we rather assume that overall the recording was critically poor before the pandemic as well? The text states ka that the values from Visit 1 come from a pre-study and thus from the time before the pandemic.

Author comments: Thank you – you are right. Visit 1 was pre-pandemic – now changed.

  1. Line 248: The explanation of the abbreviations from Table 2 should also be under Figure 1, as the figures should be understandable by themselves.

Author comments: Thank you – now done.

  1. Line 279/280: The sentence: "Common complaints among the patients, when recorded, were documented via the sentimental approach using machine learning" needs explanation. What exactly is meant by 'sentimental approach using machine learning'? Please explain which parameters were recorded or evaluated in this regard.

Author comments: Thank you for this. Here is the response. Sentiment analysis, also known as opinion mining, is a technique in natural language processing that involves using machine learning algorithms to identify and extract subjective information from textual data (Common complaints from patients). Specifically, it involves analysing text to determine the emotional tone (diseases) or sentiment.  Machine learning is used in sentiment analysis to train algorithms to recognize patterns in a language that indicate a particular sentiment. The process usually involves two phases: training and testing. In the training phase, a dataset of labelled text is used to train the algorithm to recognize specific sentiment patterns. In the testing phase, the algorithm is applied to new, unlabelled text data, and its performance is evaluated based on how well it can correctly identify the sentiment of the text. We have added some comments on this in the updated text, and hope this is now acceptable.

  1. Line 326 better: Given the extent of missing data in the patients' records, it is difficult to give further details and interpretations on this.

Author comments: Thank you – Whilst we agree with you, we believe we can make some inferences from the data provided to help guide key stakeholders in Bangladesh and wider in the future given the paucity of data in this area. Alongside this, the growing importance of managing these patients well given increasing prevalence rates in Bangladesh. We hope this is now acceptable to you.

  1. Line 374 ff: As a reviewer, I rather believe that no guidelines should be developed for monitoring diabetes during a pandemic, but that the measures taken during the pandemic should now be critically evaluated, as there must be reasonable doubt that the measures taken were all reasonable, appropriate and effective.

Author comments: Thank you – we will add this in – this also ties in with your previous comment. We hope this is now acceptable.

Round 2

Reviewer 2 Report (Previous Reviewer 2)

Authors answered all the points mentioned. However the problem of medical history still exists. The excuse that patients are too many is not acceptable. Treatment is very important for patients with diabetes. Otherwise is missing data. I think authors could not use limited time for excuse. These patients should be excluded. In addition, authors should explain if this situation was only during the pandemic or is the common clinical practice

Author Response

 Review Report (Reviewer 2)

Open Review

Quality of English Language

( ) English very difficult to understand/incomprehensible
( ) Extensive editing of English language and style required
( ) Moderate English changes required
(x) English language and style are fine/minor spell check required
( ) I am not qualified to assess the quality of English in this paper

Yes

Can be improved

Must be improved

Not applicable

Does the introduction provide sufficient background and include all relevant references?

(x)

( )

( )

( )

Are all the cited references relevant to the research?

(x)

( )

( )

( )

Is the research design appropriate?

( )

(x)

( )

( )

Are the methods adequately described?

( )

(x)

( )

( )

Are the results clearly presented?

(x)

( )

( )

( )

Are the conclusions supported by the results?

( )

(x)

( )

( )

Comments and Suggestions for Authors

Authors answered all the points mentioned. However the problem of medical history still exists. The excuse that patients are too many is not acceptable. Treatment is very important for patients with diabetes. Otherwise is missing data. I think authors could not use limited time for excuse. These patients should be excluded. In addition, authors should explain if this situation was only during the pandemic or is the common clinical practice

Submission Date

17 March 2023

Date of this review

07 Apr 2023 23:37:09

April 11, 2023

Our Answer

Respected reviewer our objective is to demonstrate the influence of the COVID-19 lockdown on health outcomes. Although we acknowledge that missing data exist in the unrecorded observations, we have found that they have a significant impact during the lockdown period. During the initial visits (Visit 1-3), the number of unrecorded observations was higher, but it decreased significantly as the pandemic situation improved. We have presented statistical comparisons in Table 3 and Figure 1, where all the unrecorded observations were treated as missing data and comparison was made only based on available data. However, in other analyses, we have reported the number or frequency of unrecorded observations.

Yes, respected reviewer you are very right that large number of patient may not be an excuse for not record keeping.  The reasons that we suspect behind non-recorded data is just our assumption on the basis of that, in this country for every 2500 patient, there is only one doctor! We didn’t focus on causes, but we tried to find out the proportion of non-recorded data. May be its a common clinical practice as you said, which is actually an another area of research interest.

Additionally, as Bangladesh yet not entered in digital record keeping age. Thereby, poor practice exists definitely both patient and physician. Furthermore, manual data keeping unable to ensure appropriate record keeping.

This manuscript is a resubmission of an earlier submission. The following is a list of the peer review reports and author responses from that submission.

Round 1

Reviewer 1 Report

This study describes the adherence to follow-up and the variations in blood tests in a group of diabetic patients before, during and after the COVID-19 pandemic in Bangladesh.

The topic is interesting but the study appears to have been difficult due to the fact that no hospital databases were available and the patients kept their notes with them and these were not always available.

In spite of this difficulty, the article presents some results which are worthy to be considered. However there are many points which need to be amended.

In detail:

Title - A shorter title would be preferable.

Abstract - line 35: BP; please use full length terms before abbreviations every first time. Check this throughout the entire manuscript.

Keywords  4 or 5 are enough.

Introduction

-          Lines 46-54 should be omitted.

-          The concept that prevalence of T2DM is increasing in Bangladesh should be elucidated only once and not repeated unnecessarily throughout the introduction (see for example line 111 and line 119)

Materials and Methods

-          Line 138: “December 2021, i.e. before the start…” I guess the authors referred to December 2019 and not 2021

-          Line 154: “In addition, the extent of any co-morbidities”. This phrase does not make any sense and should be removed.

-          Table 1: this could probably be better placed in the supplementary material section.

Results

-          Tables: there are too many and this causes a bit of confusion.  Moreover, if I understand well,  the most important data appears to be the number and percentage of NR, since this impacts on the percentages of Y and N. I suggest to put tables from 5 to 11 in the supplementary material section.

-          Table 4: it is strange that at Visit 3 both BMI and FPG have dramatically decreased: this needs an explanation

-          Table 6 – Legend: Details of insulins prescribed; GLP1-RA  is not an insulin.

-          Figure 2: omit this figure, what you say in the text is enough.

Discussion

-          Lines 316-319: unclear, please rephrase.

-          Lines 374-376: these lines are not necessary. Instead a comment should be made on the bias created by the high number of NRs.

Author Response

Comments and Suggestions for Authors

This study describes the adherence to follow-up and the variations in blood tests in a group of diabetic patients before, during and after the COVID-19 pandemic in Bangladesh.

The topic is interesting but the study appears to have been difficult due to the fact that no hospital databases were available and the patients kept their notes with them and these were not always available.

In spite of this difficulty, the article presents some results which are worthy to be considered. However there are many points which need to be amended.

In detail:

 Title - A shorter title would be preferable.

Author comments: Thank you – now changed.

 Abstract - line 35: BP; please use full length terms before abbreviations every first time. Check this throughout the entire manuscript.

 Author comments; Thank you – now changed

 Keywords –  4 or 5 are enough.

 Author comments: Thank you – we would though like to keep the 6 listed if we can. We hope this is OK with you.

Introduction

-          Lines 46-54 should be omitted.

Author comments: Thank you – now removed.

-          The concept that prevalence of T2DM is increasing in Bangladesh should be elucidated only once and not repeated unnecessarily throughout the introduction (see for example line 111 and line 119)

 Author comments: Thank you – we have now addressed this and hoe this is now acceptable.

 Materials and Methods

-          Line 138: “December 2021, i.e. before the start…” I guess the authors referred to December 2019 and not 2021

Author comments: Thank you for your kind attention – now changed.

-          Line 154: “In addition, the extent of any co-morbidities”. This phrase does not make any sense and should be removed.

Author comments: Thank you – now amended. We hope this is now OK.

-          Table 1: this could probably be better placed in the supplementary material section.

 Author comments: Thank you – now moved.

 Results

-          Tables: there are too many and this causes a bit of confusion.  Moreover, if I understand well,  the most important data appears to be the number and percentage of NR, since this impacts on the percentages of Y and N. I suggest to put tables from 5 to 11 in the supplementary material section.

Author comments: Thank you – now most of the Tables have been moved to Supplementary Tables. We hope this is now acceptable.

-          Table 4: it is strange that at Visit 3 both BMI and FPG have dramatically decreased: this needs an explanation

Author comments: Thank you very much for raising this issue. We now realise there was an error in our calculations and have updated this Table. We hope this is now OK

-          Table 6 – Legend: Details of insulins prescribed; GLP1-RA is not an insulin.

Author comments: Thank you for pointing this out. We have now changed the Title of this table in the Supplementary Table and in the revised paper, and hope this is now OK.

-          Figure 2: omit this figure, what you say in the text is enough.

 Author comments: Thank you – now done.

 Discussion

-          Lines 316-319: unclear, please rephrase.

Author comments: Thank you for pointing this out – now amended.

-          Lines 374-376: these lines are not necessary. Instead a comment should be made on the bias created by the high number of NRs.

Author comments: Thank you for pointing this out. We have now amended this section, and hope this is now acceptable.

Reviewer 2 Report

Although it is an interesting topic, there many points need improvement. 

1. English language need improvement.

2. abstract should have the form "Introduction-material methods-results-conclusions"

3. In the abstract the purpose in not clear and the conclusion missing

4. Introduction should be more brief. Authors describe the situation during covid pandemic, however readers are familiar with the prohibition. Authors should write again the introduction with less description

5. the aim of the study is not clear

6. There are two many tables with many information which makes it difficult to read

7. In the discussion authors repeat the results. They should try to explain their findings and compare them with similar studies. If there are no similar studies, they should compare their findings with what happened in other pandemics or in different diseases

8. They should add inclusion and exclusion criteria

9. They should add limitations of their study

Author Response

Comments and Suggestions for Authors

Although it is an interesting topic, there many points need improvement. 

  1. English language need improvement.

Author comments: Thank you – we have now been through the document with the help of one of the co-authors who is a native English speaker with over 500 publications in peer-reviewed Journals since 2008. We hope this is now acceptable.

  1. abstract should have the form "Introduction-material methods-results-conclusions"

Author comments: Thank you for this comment. However – the instructions from the Journal ask for a continuous abstract and not to break it down with these sub-headings. We trust this is acceptable to you.

  1. In the abstract the purpose in not clear and the conclusion missing

Author comments: Thank you for pointing this out. We have now updated the abstract and hope this is now OK.

  1. Introduction should be more brief. Authors describe the situation during covid pandemic, however readers are familiar with the prohibition. Authors should write again the introduction with less description

Author comments: Thank you for your comments. We have now been through the Introduction and made it more focused. We hope this is now acceptable.

  1. the aim of the study is not clear

Author comments: Thank you for pointing this out. As stated, we have now been through the Introduction and made it more focused including the aims. We hope this is now acceptable.

  1. There are two many tables with many information which makes it difficult to read

Author comments: Thank you. As seen, we have now moved most of the Tables to Supplementary Tables to make the results more focused. We hope this is now OK.

  1. In the discussion authors repeat the results. They should try to explain their findings and compare them with similar studies. If there are no similar studies, they should compare their findings with what happened in other pandemics or in different diseases

Author comments: Thank you – we have now consolidated some of the findings as well as made reference to other NCDs/ other LMICs. We hope this is now OK.

  1. They should add inclusion and exclusion criteria

Author comments: Thank you – now made clearer in the revised paper.

  1. They should add limitations of their study

Author comments: Thank you – this appears at the end of the Discussion. We have now added to expanded on this, and hope this is now OK.

Reviewer 3 Report

The paper takes up a very important question of how chronic disease care was under the pandemic conditions with lockdown in Bangladesh using T2DM as an example.
Unfortunately, the work has serious implausibilities. In Table 4, blood glucose levels (FPG and PPG) are reported in mmol/L. The mean of 40.21 with a standard deviation of 61.6 seems implausible, as do the other blood glucose values at visits 1 to 5. Are incorrect units given here? Is there a conversion factor error?
The BMI in visit 3 with a value of 16.22 also seems implausible compared to the values from visits 1, 2, 4 and 5.
The unit of measurement for the serum creatinine is completely missing (probably mg/dl). Tables 5 to 11 do not make sense if the values for the individual parameters are mostly missing by up to 90%. 
Furthermore, there are many corrections and errors in the text (e.g. lines 46 to 54 are probably a note that is not meant for publication, etc.).
Despite the important question also for the country Bangladesh, the work cannot be scientifically accepted in this form from my review point of view, but needs a thorough revision and recalculation of the table values (especially in Table 4).

Author Response

Comments and Suggestions for Authors

1) The paper takes up a very important question of how chronic disease care was under the pandemic conditions with lockdown in Bangladesh using T2DM as an example.

Author comments: Thank you for this positive comment

2) Unfortunately, the work has serious implausibilities. In Table 4, blood glucose levels (FPG and PPG) are reported in mmol/L. The mean of 40.21 with a standard deviation of 61.6 seems implausible, as do the other blood glucose values at visits 1 to 5. Are incorrect units given here? Is there a conversion factor error?

Author comment: We thanked to the reviewer for pointing out this concern. Further analysis showed some errors found in the data entry, which we have now corrected. Besides this, some outliers have also been removed from the FPG and PPG values. After making these corrections, we have re-analysed the data and presented this in the updated Table 3. We hope this is now OK

3) The BMI in visit 3 with a value of 16.22 also seems implausible compared to the values from visits 1, 2, 4 and 5.

Author comment: Thank you – this was an initial error which has now been corrected. We hope this is now acceptable.

4) The unit of measurement for the serum creatinine is completely missing (probably mg/dl).

Author comment: Thank you – now amended in the updated Table in Supplementary material.

5) Tables 5 to 11 do not make sense if the values for the individual parameters are mostly missing by up to 90%. 

Author comment: Thank you for this. We have now made comments in the Methods section about missing data – and acknowledged this as a limitation. We hope this is now acceptable as this reflects the situation that did occur in Bangladesh pre- and during the pandemic, which we are now looking to build upon. We hope this is acceptable to you.

6) Furthermore, there are many corrections and errors in the text (e.g. lines 46 to 54 are probably a note that is not meant for publication, etc.).

Author comment: Thank you for this. We have also now been through the document with the help of one of the co-authors who is a native English speaker with over 500 publications in peer-reviewed Journals since 2008. We hope this is now acceptable.

7) Despite the important question also for the country Bangladesh, the work cannot be scientifically accepted in this form from my review point of view, but needs a thorough revision and recalculation of the table values (especially in Table 4).

Author comment: Thank you. As mentioned, we have now updated Table 3 (old Table 4) and corrected the errors. In addition – we believe this paper has pointed out serious concerns in Bangladesh regarding the situation that did exist regarding the management of patients with T2DM across the sectors (private and public) that is similar to a number of other LMICs with similar paper-based systems. This needs to be urgently addressed (alongside improve detection and management of these patients generally) given rising rates of T2DM and associated complications across LMICs which we have pointed out. Consequently, we hope our revisions are acceptable and this paper can eventually be published in this Journal.
